# Keypoint Detection for Injury Identification during Turkey Husbandry Using Neural Networks

**DOI:** 10.3390/s22145188

**Published:** 2022-07-11

**Authors:** Nina Volkmann, Claudius Zelenka, Archana Malavalli Devaraju, Johannes Brünger, Jenny Stracke, Birgit Spindler, Nicole Kemper, Reinhard Koch

**Affiliations:** 1Science and Innovation for Sustainable Poultry Production (WING), University of Veterinary Medicine Hannover, Foundation, 49377 Vechta, Germany; 2Institute for Animal Hygiene, Animal Welfare and Animal Behavior, University of Veterinary Medicine Hannover, Foundation, 30173 Hannover, Germany; birgit.spindler@tiho-hannover.de (B.S.); nicole.kemper@tiho-hannover.de (N.K.); 3Department of Computer Science, Faculty of Engineering, Christian-Albrechts-University, 24118 Kiel, Germany; cze@informatik.uni-kiel.de (C.Z.); archanamd1996@gmail.com (A.M.D.); jobr@informatik.uni-kiel.de (J.B.); rk@informatik.uni-kiel.de (R.K.); 4Institute of Animal Science, Farm Animal Ethology, University of Bonn, 53115 Bonn, Germany; jenny.stracke@itw.uni-bonn.de

**Keywords:** turkeys, keypoint detection, crowded dataset, pose estimation, injury location, animal welfare

## Abstract

Injurious pecking against conspecifics is a serious problem in turkey husbandry. Bloody injuries act as a trigger mechanism to induce further pecking, and timely detection and intervention can prevent massive animal welfare impairments and costly losses. Thus, the overarching aim is to develop a camera-based system to monitor the flock and detect injuries using neural networks. In a preliminary study, images of turkeys were annotated by labelling potential injuries. These were used to train a network for injury detection. Here, we applied a keypoint detection model to provide more information on animal position and indicate injury location. Therefore, seven turkey keypoints were defined, and 244 images (showing 7660 birds) were manually annotated. Two state-of-the-art approaches for pose estimation were adjusted, and their results were compared. Subsequently, a better keypoint detection model (HRNet-W48) was combined with the segmentation model for injury detection. For example, individual injuries were classified using “near tail” or “near head” labels. Summarizing, the keypoint detection showed good results and could clearly differentiate between individual animals even in crowded situations.

## 1. Introduction

Research on farm animal welfare and behavior now utilizes computer vision and deep learning technologies. In the best-case scenario, such approaches can support, simplify, and, above all, accelerate continuous animal observation. Furthermore, implemented real-time monitoring of large animal flocks such as in conventional poultry farming that uses computer vision and machine-learning algorithms can prevent large-scale outbreaks of diseases or behavioral disorders [1]. For example, previous studies regarding poultry farming evaluated behavior [2], lameness [3], feeding [4,5], lighting preferences [6], or movement [7,8] based on new PLF technologies.

Analyzing animal behavior and health should be conducted with minimal human interference and involvement to not unnecessarily affect animals or disturb their natural behavior. Computer vision is a proven and non-invasive technology for video and image data collection [9]. Computer vision tasks can use pose estimation, which provides important behavioral information. Pose estimation can be described as follows: individual objects are abstracted into keypoints, i.e., spatial locations of interest such as body parts or joints. These keypoints are built into skeletons, and poses are finally estimated on them. To enhance the recognition precision, additional markers can be placed on the studied animal, although this method could distract it and could be very expensive depending on the number of individuals [10]. Alternatively, modern approaches for pose estimation of animals are supported by non-invasive vision-based solutions such as keypoint detection (KPD). Thus, keypoints are marked manually on sample images or video frames to form a skeleton model to record an individual animal as well as estimate its pose [11,12].

In turkey husbandry, injurious pecking against conspecifics is a widespread and serious problem in animal welfare [13]. The early detection of the occurrence of this injurious pecking in a turkey flock can avoid serious wounding. Indeed, bloody injuries trigger further pecking behavior [14], and thus an early intervention can prevent an outbreak of this behavioral disorder [15]. One option to support the turkey farmer in monitoring the flock with regard to animal welfare-related issues such as the occurrence of injurious pecking is the use of computer vision systems. In a preliminary study, the foundations were laid for the development of an image-based automated system using a neural network to detect pecking injuries in a turkey flock [16]. A neural network was trained based on the manual annotations of (color) alterations to skin and plumage on images of turkey hens. Various additional work steps were then performed to improve the detection assessment. However, the essential issue in the preliminary study was uncertainty regarding the correct evaluation. This primarily occurred in the case of plumage injuries in which detection was difficult due to shadows, turkeys’ posture, and/or overlapping of the individual animals. In the system developed to date, there was also an increased rate of false positives due to erroneously detected ‘injuries’ on the litter or on stable equipment. To tackle these problems and reduce the false-positive detections in further research, the present study aimed to provide more information and therefore first to train the network in identifying the animal and its body regions (e.g., head, neck, back, tail, and wings) in order to then, in a second step, detect potential injuries on the previously identified animal body.

Thus, the aim of this study was to compare different models to find the most suitable model for KPD in fattening turkey hens recorded with top-view cameras. Furthermore, we strived to connect two detection systems to provide more information on the animal position and subsequently predicted an injured location.

## 2. Related Work

Analyzing animal behavior via tracking and monitoring has been implemented by different tools including radio-frequency identification (RFID) transponders [17], accelerometers [18], and cameras coupled with image analysis [19]. Videos or images have been analyzed and used for studies on broilers including bodyweight [20,21], health status [22,23], behavior [24], flock movement [25], and locomotion/activity [3,26]. The technology has also been used in the poultry sector for research into butchering [27], carcass and meat monitoring [28,29], and egg quality analysis [30,31].

A recent review on tracking systems for the assessment of farmed poultry stated that computer vison systems can be used for a variety of applications such as checking images for the presence of poultry, classifying the identified animal as sick or absent, determining feeding and watering structures, or for locating the exact position of poultry in an image [32]. The so-called keypoints can offer more detailed information about the body and body parts of a recorded animal. KPD algorithms can locate these areas in isolation, and pose estimation can detect these keypoints and connect their structural information [33]. Thus, these pose estimation models have mainly been used for humans [34,35]. They were also tested on laboratory animals when recorded in a controlled environment, e.g., mice [36], locusts [37], fruit flies [38], and even worms (*C. elegans*) [39]. However, there are relatively few architectures for recognizing the poses of farm animals such as cows [33,40,41], pigs [42], and broiler chickens [10]. By detecting the different body keypoints and their locations, these tools can offer activity recognition or video surveillance in humans.

Fang et al. [10] combined a pose estimation network of broiler chickens with a classification network to analyze broiler chickens’ behavior. They used the front view of a broiler’s head and the side view of the body to construct a pose skeleton through the feature points of a chicken and then tracked specific body parts in various behaviors such as eating, resting, or running. Finally, they stated that their research provided an appropriate non-invasive method to analyze chicken behavior. More recently, Doornweerd et al. [43] reported the performance of an animal pose estimation network: the network was investigated, trained, and tested on multi-species data from broilers as well as from turkeys. They collected data for pose and gait assessments and evaluated a multi-species model to reduce the required dataset and finally annotation needs. Doornweerd et al. [43] recorded the turkeys walking along a corridor, and they used a view from behind the animals, paying particular attention to their locomotion system. They then defined eight key points (head, neck, right and left knee, right and left hock, and right and left foot).

In contrast to previous pose estimation studies on poultry, this study focuses on injury detection in turkeys, which requires a complete view of the animals (not only a lateral view). Therefore, this method offers a KPD method on images from top-view cameras. This paper further proposes to combine a KPD model with a segmentation model to localize injuries.

## 3. Materials and Methods

### 3.1. Preliminary Research

The dataset of turkey images used here originates from a previously described study that detected pecking injuries in a turkey flock on a German research farm using neural networks [16]. Three top-view video cameras (AXIS M1125-E IP-camera, Axis Communications AB, Lund, Sweden) were installed ~3.0 m above the ground to capture the top-view videos of the animals. The turkey hens (n = 2170, B.U.T. 6, Aviagen Group, Newbridge, UK) were observed during their entire fattening period of 16 weeks; two periods were analyzed. The recordings were at different locations in the barn and at different dates; therefore, they were used at different ages of the birds. The images were of great diversity and contained diverse environmental impacts such as drinkers, feeders, and/or litter material (Figure 1). The turkeys were present in a crowded state. Sometimes certain body parts were hidden or even missed. Therefore, the data set was considered to be highly diverse and difficult for model predictions.

Our preliminary research developed software [16] to mark the injuries visible on the images by human observers. The software consisted of a web application front end and server back end; it allowed multiple annotators to work together to produce a dataset with low intra- and inter-observer variance. A neural network was later trained with these annotations to learn to detect such injuries on other unknown images from the same domain. Due to unacceptable agreement between the annotations of humans and the network, several work steps were initiated to improve the training data and thus the performance of the network [16]. Finally, the different work steps involved could be viewed as meaningful even if the system itself still required further improvements.

### 3.2. Manual Keypoint Annotation

The images of the animals were recorded via top-view cameras, and the turkey keypoints were defined by seven points visible from the top and shown in Figure 2a,b.

The turkey data set was manually annotated using an annotation software tool called Supervisely (San Jose, CA, USA)—a web platform for computer vision developed by Deep Systems (Moscow, Russia). Overall, 244 images showing different situations, compartments, age groups, and stocking densities were marked. The total number of annotated individual animals was 7660 turkey hens. Supervisely annotations were stored in a JavaScript Object Notation (JSON) format, and the keypoints were converted into the standard Common Objects in Context (COCO) JSON [44] using Python (Python Software Foundation, Python Language Reference, version 3.8. available at http://www.python.org, accessed on 13 August 2021). The COCO JSON format is a single file containing annotations of all training images; it can be directly used to train a model or be converted to other standard formats.

After manually annotating the keypoints, a bounding box was generated around each turkey hen via a tight bounding rectangle of all seven keypoints. This was then saved in the dataset. The area of the bounding box was calculated by its length and width and then saved as the segmentation area value. For visualization, the images are overlaid with their corresponding annotation as shown in Figure 3.

### 3.3. Keypoint Detection Models

Two different state-of-the-art deep learning algorithms for KPD were evaluated. The “Baseline for Human Pose Estimation” by Xiao et al. [45] provides a good baseline and high speed. A more intricate approach is “High-Resolution Representation for Object Detection” (HRNet) [46].

The first step in the evaluation of keypoint estimation networks by Xiao et al. [45] is to apply a backbone network on the input image to generate the network activations. The so-called feature maps mark a lower dimensional response to the network. Therefore, the ResNet (Residual Neural Network) architecture for all backbone networks was chosen. The ResNet model is one of the most popular and successful deep learning models as well as the most common backbone network for image feature extraction [45]. It was designed to address the problem of decreasing accuracy. It increases the depth of the neural networks [47]. The ResNet uses skip connections, and its architecture tries to model the residual of the intermediate output instead of the traditional method of modeling the intermediate output itself. The baseline KPD directly feeds the resulting features into a deconvolution module to predict a mask for every keypoint. In these masks, the keypoint locations are marked with high values (with a 2D-Gaussian blur) for the predicted position that is low or close to zero values at all other positions.

The ResNet backbone network was initialized by pre-training on the ImageNet approach proposed by Deng et al. [48] as a classification dataset. Here, ResNet was used at different depths with 50, 101, and 152 layers. This pre-training taught the network to be aware of common image features that signify objects (such as edges) before being applied to the turkey images; thus, the amount of training data required was drastically reduced. For training, the difference between the generated mask from the network and the target mask from the annotation were compared in the loss function; the network was adjusted using standard back-propagation.

The HRNet is a general-purpose convolutional neural network for tasks such as semantic segmentation, object detection, and image classification. Wang et al. [49] stated that the three fundamental differences from existing low-resolution classification networks of high-resolution representation learning networks are that they:(i)Connect high- and low-resolution convolutions in parallel rather than in series;(ii)Maintain high resolution through the entire process instead of recovering high resolution from low resolution; and(iii)Fuse multi-resolution representations repeatedly, thus rendering rich high-resolution representations with strong position sensitivity.

An overview of the HRNet architecture is shown in Figure 4.

Here, the HRNet-W48 (big size) and the HRNet-W32 (small size) were evaluated where 32 and 48 represented the widths of the high-resolution subnetworks. The key difference to the baseline—which only uses the output of a backbone network—is that HRNet can maintain high-resolution representation throughout the process.

The largest impact on model performance was shown by the hyper-parameter learning rate. The learning rate parameter was evaluated using three different settings between 1e−4 and 5e−4, and the model performance was evaluated for each setting every ten training epochs. These performances are listed in Table 1.

The implementation of both methods was based on the OpenMMLAb Pose Estimation Toolbox (available at https://github.com/open-mmlab/mmpose, accessed on 1 September 2021) and tested on benchmarks of the COCO KPD dataset [44].

The standard evaluation metric was based on Object Keypoint Similarity (OKS) according to the COCO evaluation metric (see http://cocodataset.org/#keypoints-eval, accessed on 13 August 2021) and was used to quantify the closeness of the predicted keypoint location to ground truth keypoints on a scale between 0.0 and 1.0 (see Equation (1)).
(1)$$OKS=\frac{\Sigma_{i}\left[exp\left(-\frac{di2}{2s2{ĸ}_{i}2}\right)\delta \left(\nu_{i}>0\right)\right]}{\Sigma_{i}\left[\delta \left(v_{i}>0\right)\right]}$$

Here, di2 is the Euclidean distance between the detected keypoint and the corresponding ground truth, *ν_i_* is the visibility flag of the ground truth, *s* is the object scale, and *ĸ_i_* is a per-keypoint constant that controls falloff.

An OKS-threshold classified whether a keypoint location was correct or not. Only a few points will be detected upon choosing a high value: these have a high certainty or, in statistical terms, they have a high precision. The opposite is seen upon choosing lower values: there are more detected points and a higher recall; thus, there is a higher ratio of points that will be identified correctly. This was obviously a trade-off, and thus both results for different threshold values were noted. The KPD evaluation was performed with 0.50 (loose metric) and 0.75 (strict metric) as reported thresholds. We evaluated the average precision at these thresholds as AP_50_ and AP_75_ as well as the average recall (AR_50_, AR_75_). The average precision without a named threshold AP is a more abstract measure and averages over different OKS thresholds between 0.50 and 0.95. This strategy offers a combined view: AP = the mean of AP scores at 10 positions, OKS = 0.50, 0.55, …, 0.90, 0.95. The average recall without a named threshold AR is the analogue measure for the recall: AR = the mean of AR scores at 10 positions, OKS = 0.50, 0.55, …, 0.90, 0.95.

### 3.4. Segmentation Model

As described in the previous study (see Section 3.1 [16]), human observers processed the images of turkey hens and manually annotated the visible injuries. A network for semantic segmentation was then trained with these annotations. This U-Net was based on an efficient net backbone [50,51] and is a convolutional network architecture for fast and precise segmentation of images. It can localize and distinguish borders by performing classification on every pixel, so that the input and output share the same size [50]. Thus, pixelwise masks of injuries were generated building on this previous work.

### 3.5. Combination of Models

The next step combined the evaluated KPD models and the segmentation model discussed in the last section. First, the keypoints were detected and mapped to the original image to preserve the original scaling. A segmentation model for injuries was then applied and rescaled, accordingly. Before injury segmentation, we added several post-processing steps to the keypoint output image. The closet keypoint was noted for every injury, and thus any detection was identified as one of the following injuries: beak (B), head (H), neck (N), left wing (L), right wing (R), center of the body (C), and tail (T). This classification followed the keypoint schema shown in Figure 2a,b. If no closest keypoint was found, then the “related” injury was identified as a false positive segmentation; thus, the accuracy of the total assessment should have increased. Finally, the keypoint information could be harnessed to find pecking injuries in the turkey hens using this combination of data deepened in further studies.

## 4. Results

### 4.1. Quantitative Results

The three different versions of the ResNet backbone provided an increasing number of layers, which implies higher runtime and more complexity, but probably an increased performance up to a certain point. Starting with the baseline method and a backbone with 50 layers of ResNet architecture backbone, the number of layers steadily increased to 101 layer steps and finally became very high (152 layers). However, HRNet outperformed the 152 backbone layers in the baseline method with only 32 layers. This could be increased even more with the 48 layers of the HRNet-W48 network architecture. The results of the OKS metrics for the KPD model evaluations are listed in Table 2.

Our evaluations showed that the best model performance of HRNet-W48 was observed with a learning rate of 5e−4, a batch size of 64 (limited by available GPU memory), and 100 epochs of training (Table 2). Longer training resulted in overfitting for which we recommend the periodic evaluation of model performance or early stopping. HRNet led to better quantitative results than the baseline KPD models even with a sub-optimal learning rate.

### 4.2. Qualitative Results

An evaluation set of images was withheld during the training of KPD. Here, a baseline method with 152 layers and an HRNet with 48 layers were tested. The qualitative results are shown in Figure 5.

In the example image (Figure 5), clear differences were visible between the KPD results of the baseline method and HRNet-W48. The HRNet showed better results in most cases. Hence, we used the previously identified HRNet-W48 for the next evaluation because it was quantitatively and qualitatively best (see Table 1), i.e., the combination of KPD with injury detection was based on a segmentation model from previous work [16].

A representative image of the combination of detection models is shown in Figure 6. The turkey hens on this image are very close to each other, and the classification of the individual injury was challenging. Thus, the detected locations were noted using labels such as “near neck”, “near beak”, or “near tail” (Figure 6).

## 5. Discussion and Further Work

Keypoint detection (KPD) and pose estimation are non-invasive methods to predict animal location on videos or images. They can be used to define body parts or to analyze animal health and behavior. Here, two state-of-the-art approaches for KPD on turkey hens were adjusted and evaluated. A better KPD method was then combined with a segmentation model to detect injuries to present to the corresponding injury locations.

Generally, image acquisition using computer vision technology in poultry housings is realized either by top-view or side-view camera position [52]. Previously, different authors used top-view camera imaging to measure broilers’ weight [20,21], to analyze the distribution of the animals in the barn [53], or to detect sick birds [22]. Our work used computer vision technology and showed that KPD based on top-view turkey images was possible. The detection could differentiate between individual animals even in the crowded situations seen in conventional poultry housing where several thousand animals are kept together in a flock. Some only partially visible turkey hens on the image border were missed with KPD, but this was not detrimental due to our unique use case.

The results of a quantitative evaluation of the two performed KPD models showed that higher network complexity led to better results and thus better performance of the HRNet-W48 model. The qualitative differences between baseline and HRNet seen in the images were thus confirmed with the OKS metrics. High values for the loose metric AP_50_ of up to 0.74 were reached—these are similar to results reported by other authors in challenging situations [46]. The values for the AP_75_ strict metric and also the AP, which includes an even stricter threshold, were lower and showed a less accurate result for some keypoints. The exact location of some keypoints (e.g., left or right wing) were only roughly visible in top-down view especially when the animals had moved, were grooming themselves, or were sleeping in a different posture. The detection accuracy of these keypoints was limited. Such conditions were common in the recordings of turkeys’ natural behavior in the flock, and thus this could explain the reduced values of AP_75_ and AP.

One limitation of our top-down view camera-based approach is that certain affected regions, such as the cloaca or the lateral side of the wings, are hard to capture. Besides that, the annotation of keypoints on the all-white bodies of the turkeys was already difficult, and thus we could not guarantee that, for instance, the “center of the body” keypoint always had the same position. Doornweerd et al. [43] estimated turkey poses based on keypoints placed on hocks and feet. Such keypoints can probably be defined more precisely than on images from a top-view camera. Fang et al. [10] described the center of the body of a chicken in more detail by setting the center of the largest inscribed circle in the contour boundary of the broiler chicken. They used the front view of broiler chicken heads as well as a side view and specified keypoints such as the eyes, the highest point of the comb, or the tip of the beak. Nevertheless, the animals were recorded in the barn in this study and thus in their usual environment. The turkeys were neither stimulated to walk along a corridor [43] nor placed in a specific photography environment [1] as in other studies.

The system for injury detection we are creating will be applied in farm settings; thus, we accepted a loss in model performance. The recordings from the barn were also characterized by an uncontrolled environment with a non-uniform background and varying light conditions. Furthermore, as mentioned above, the turkeys were frequently occluded by objects on the farm or by other animals. This can lead to inaccurate pose estimates as well [41]. Thus, every individual was annotated to prevent the model from learning conflicting information. Such an annotation assessment was even more laborious and time-consuming than annotations for individual pose estimations.

For more information on the injuries detected in the preliminary study [16], we combined the U-Net segmentation model for injury detection with the better-performing KPD from this study. Therefore, the HRNet-W48 model was combined to classify injury locations. First, exemplary results of injury localization were presented using labels such as “near neck”, “near beak”, or “near tail”. Finally, for the system to be developed, the precise location of an injury will not be critical or important. Rather, an increase in the rate of injuries in the turkey flock would trigger an alarm. In this case, the definition of the location of an injury should verify its potential. As in the preliminary study [16], some injuries were false positives because they were localized in the litter, on feeders, or were simply other objects in the compartments. A localization of injuries could therefore be used to ignore such illogical events in further development. Previous studies have shown that injuries to the head, neck, and back occur more frequently [54], and thus the detection of the body regions by KPD could also be used for further verification of existing injuries.

There is still no individual animal identification system for poultry in contrast to cattle. Individual tracking is nearly impossible in crowded housing situations such as a turkey barn [16]. To detect the density in a poultry flock, Cao et al. counted the chickens in an image using point supervision [55]. Counting of turkeys in the barn, which were previously detected using KPD, would also be conceivable for further research on our system to be developed, as in this manner thresholds for injuries’ frequency related to the admitted turkeys would be possible.

Further research is needed to ensure that the use of KPD can improve the accuracy of an injury detection system to be developed. Obviously, the overarching aim remains a system to monitor the turkey flock for animal welfare and to reduce financial losses. Such a system can draw attention to existing pecking injuries to enable intervention and the separation of the injured animal.

## 6. Conclusions

This paper proposed and evaluated different keypoint detection (KPD) models on images recorded in a turkey hen flock where the partially crowded animal behavior led to overlapping on the images. Overall, the use of KPD in turkey hens showed good results, and the HRNet-W48 model provided the best performance. Therefore, in a first attempt for injury localization, the HRNet-W48 model was combined with an injury detection model (resulting from a preliminary study). In future work, the classification of individual injuries as “near tail” or “near left wing” could include a plausibility check. Therefore, such injury localization could improve the accuracy of automatic injury detection in the turkey barn.

## Figures and Tables

**Figure 1 sensors-22-05188-f001:**
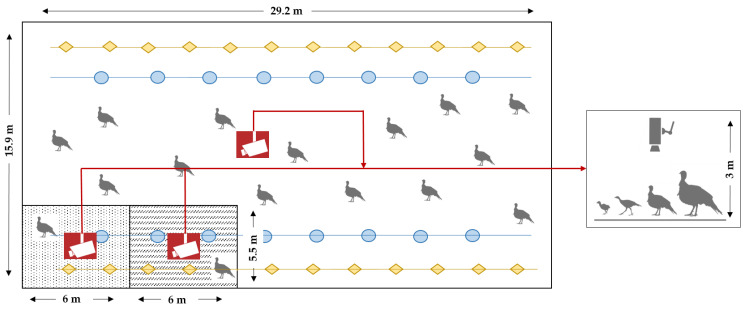
Schematic view of the turkey barn (15.9 × 29.2 m) showing the different positions of the three top-view video cameras. The feeding lines are shown with orange squares, and the drinking lines have blue circles. A separate experimental compartment (5.5 × 6 m) and a second compartment (5.5 × 6 m) for sick animals are shown as differently patterned squares.

**Figure 2 sensors-22-05188-f002:**
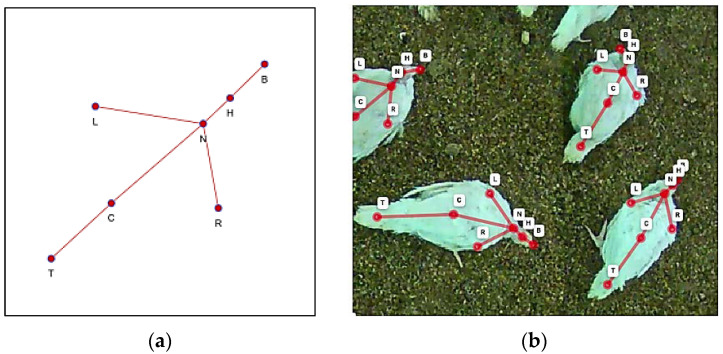
(**a**) Keypoint skeleton showing the beak (B), head (H), neck (N), left wing (L), right wing (R), center of the body (C), and tail (T). (**b**) Example image showing the keypoints on turkey hens.

**Figure 3 sensors-22-05188-f003:**
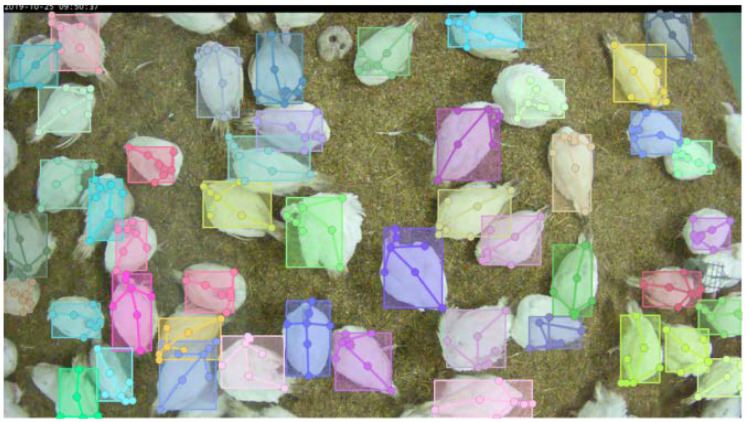
Example image showing the visualization of annotated keypoints and bounding boxes using the COCO API [44].

**Figure 4 sensors-22-05188-f004:**
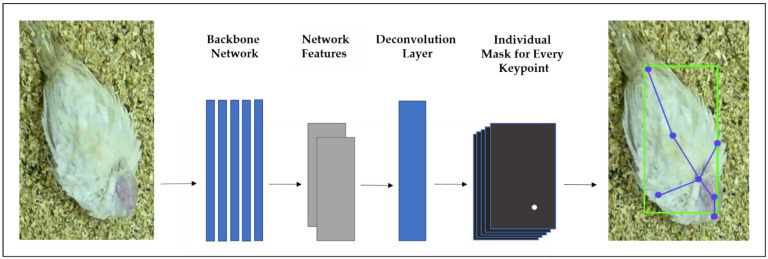
Overview of the baseline keypoint detection method by Xiao et al. [45].

**Figure 5 sensors-22-05188-f005:**
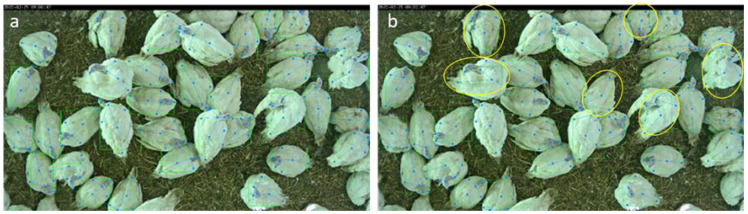
Comparison of KPD using (**a**) baseline method with 152 layers and (**b**) HRNet-W48. Turkeys showing differences between the results of the baseline and HRNet are highlighted with yellow circles on the right image.

**Figure 6 sensors-22-05188-f006:**
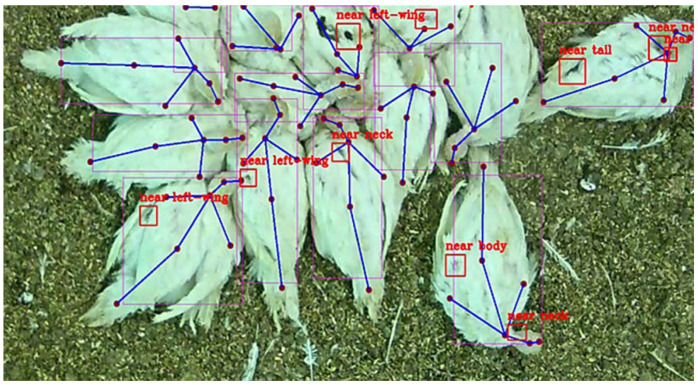
Combination of KPD generated in this study and injury detection from previous work [16] on the evaluation dataset. The keypoints are shown in lilac connected by blue lines. The supposed injuries detected are highlighted using red boxes and the classification of the injuries is marked using labels such as “near neck” or “near tail”.

**Table 1 sensors-22-05188-t001:** Performance of the HRNet-W48 showing the used hyper-parameters for model training and structure. Object keypoint similarity (OKS) metrics state the average precision with threshold values of 0.50 (AP_50_) and 0.75 (AP_75_); these were averaged over thresholds from 0.5 to 0.95 (AP) as well as the average recall with threshold values of 0.50 (AR_50_) and 0.75 (AR_75_) averaged over thresholds from 0.50 to 0.95 (AR). We used a batch size of 64 for all tests. Best-performing values are printed in bold. We evaluated the model performance every 10 epochs to select the best-performing model and then listed the performance for this epoch.

Hyper-Parameters	AP_0.50_	AP_0.75_	AP	AR_0.50_	AR_0.75_	AR
LR ^1^ = 1e–4; epochs = 180	0.677	0.129	0.249	0.721	0.234	0.315
LR ^1^ = 3e–4; epochs = 150	0.714	0.137	0.273	0.755	0.243	0.334
**LR ^1^ = 5e–4; epochs = 100**	**0.735**	**0.246**	**0.322**	**0.762**	**0.355**	**0.383**

^1^ Learning rate.

**Table 2 sensors-22-05188-t002:** Object keypoint similarity metrics (OKS) resulting from the different keypoint detection models stating the average precision with threshold values of 0.50 (AP_50_) and 0.75 (AP_75_) and averaged over thresholds from 0.5 to 0.95 (AP) as well as the average recall with the threshold values of 0.50 (AR_50_) and 0.75 (AR_75_). These were averaged over thresholds from 0.50 to 0.95 (AR). Best-performing values are printed in bold.

Architecture Type	AP_0.50_	AP_0.75_	AP	AR_0.50_	AR_0.75_	AR
Baseline–ResNet50	0.648	0.107	0.213	0.691	0.198	0.292
Baseline–ResNet101	0.640	0.107	0.228	0.687	0.200	0.288
Baseline–ResNet152	0.659	0.134	0.254	0.703	0.231	0.313
HRNet-W32	0.692	0.158	0.267	0.726	0.241	0.323
HRNet-W48	**0.735**	**0.246**	**0.322**	**0.762**	**0.355**	**0.383**

## Data Availability

The datasets used and/or analyzed during the current study are available from the corresponding author on reasonable request.

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
