# Peer review of "Keypoint Detection for Injury Identification during Turkey Husbandry Using Neural Networks"

_sensors, 2022, doi:10.3390/s22145188_

Round 1
Reviewer 1 Report
The manuscript entitled: “Keypoint detection for injury identification during turkey husbandry using neural networks” proposed an interesting and updated approach to the injuties identification in turkey husbandry. There are a few comments to address, in particular the Abstract mentions that “The injury location will be improved in future work”: please comment on this: is it necessary further work? In this case please exploit bettert he limits and end points of the proposed study. Onsider alsothat at line 131-132it is statedthat: “This paper further proposes to combine a KPD model with a segmentation model to localize injuries”. Other comments are in the following.
Lines 81-84 seem not necessary, and Section 2 could be integrated in the Introducion with proper reduction of the text.
Please add information on the mentioned “preliminary research developed software to mark the injuries visible on the images by human observers”.
The authorization and Etical Committee opinion and approval should be added for studies on animals. The images reprsent a model or real animals? Please specify.
How many animals have beenstudied? Where the study has been done? Please add information on this.
The statistical analysis should added and discussed for better assessment of the reported results.
Why the quantitative results are reported before qualitative results: please comment on this.
The end points and limits of the proposed manuscript should be better assessed and defined. I am afraid to say that the manuscript cannot be recommended for publication in the Journal.
Author Response
Dear Reviewer 1.
Thank you for your constructive comments and your helpful input.
In the following you will find a list of your comments (in italics) and our reply (changes in the manuscript are marked in red).
Since you've indicated that moderate English changes are required, we would like to inform you that previously the manuscript was edited by the expert staff of the American Manuscript Editors.
The manuscript entitled: “Keypoint detection for injury identification during turkey husbandry using neural networks” proposed an interesting and updated approach to the injuries identification in turkey husbandry. There are a few comments to address, in particular the Abstract mentions that “The injury location will be improved in future work”: please comment on this: is it necessary further work? In this case please exploit better the limits and end points of the proposed study. Consider also that at line 131-132 it is stated that: “This paper further proposes to combine a KPD model with a segmentation model to localize injuries”. Other comments are in the following.
Thank you for this important hint. This was confusing, we are sorry for that. The sentence originated from an earlier version with limited scope. Injury localization is not future work, but is actually already part of this work. This is described in line 131-132 and can be seen in Figure 6. We removed this sentence from the abstract and also changed a corresponding sentence in the conclusion to reflect that.
- Lines 81-84 seem not necessary, and Section 2 could be integrated in the Introducion with proper reduction of the text.
Thank you for your suggestions. We deleted the lines 81-84 of the Introduction. However, we would like to keep Section 2 Related Work as it could help the reader compare our research with published matter that technically relates to the proposed work. But, in order to take into account your recommendation and your valuation that the introduction "must be improved" in terms of a sufficient background and relevant references, we have shortened this part.
- Please add information on the mentioned “preliminary research developed software to mark the injuries visible on the images by human observers”.
We added the relevant citation [16] to the text and further information as follows (L155-157):
The software consisted of a web application front end and server back end, it allowed multiple annotators to work together to produce a dataset with low intra and inter observer variance.
We think, an even more detailed explanation of the previous work and the used software can be read in the cited paper.
- The authorization and Ethical Committee opinion and approval should be added for studies on animals.
The manuscript includes a paragraph regarding ethical approval (L440ff). It now reads as follows:
Institutional Review Board Statement: The experiments comply with the requirements of the ethical guidelines of the International Society of Applied Ethology [56]. All animals were housed according to EU (9) and national law. An approval by the Ethics Committee was not required for this manuscript. Before starting the study, this matter was discussed with the Animal Welfare Officer of the University of Veterinary Medicine Hannover, Foundation, Germany, and the official institution which is responsible for animal experiments in Lower Saxony, the LAVES (https://www.laves.niedersachsen.de/startseite/, accessed on 01.06.2020). All agreed on the fact that no approval was necessary, because in compliance with European Directive 2010/63/EU Article 1 (5) f), the experiments did not imply any invasive treatment, and no negative effects on the animals were expected.
- The images represent a model or real animals? Please specify.
This study was conducted on images of real animals. This is described in the section 3.1. Preliminary Research (L134ff). To clarify this even more we rewrote the following sentence (L138):
Three top view video cameras (AXIS M1125-E IP-camera, Axis Communications AB, Lund, Sweden) were installed ~ 3.0 m above the ground to capture the top-view videos of the animals.
- How many animals have been studied? Where the study has been done? Please add information on this.
The number of the studied animals can be found in line 139ff:
The turkey hens (n = 2170, B.U.T. 6, Aviagen Group, Newbridge, UK) were observed during their entire fattening period of 16 weeks; two periods were analyzed.
We added the information on where the study was done to the text. It reads (L135ff):
The dataset of turkey images used here originates from a previously described study that detected pecking injuries in a turkey flock on a German research farm using neural networks [16].
- The statistical analysis should added and discussed for better assessment of the reported results.
We think that the Object Keypoint Similary measure (Equation 1) reported in Section 3.3. Keypoint Detection Models and used in Section 4.1. Quantitative Results is a state of the art statistical measure of analysis and it is used the most prominent benchmark in this field MSCOO. As Equation 1 shows, it is built on the concepts of true positives, false positives, true negatives and false negatives and compares the result of the model with ground truth annotations.
Please note that also we report different values of this measure for four different levels of average precision and average recall in Table 1 and 2.
Therefore, we assume we explained the statistical analysis sufficiently.
- Why the quantitative results are reported before qualitative results: please comment on this.
In the quantitative results small differences are easier to see than in qualitative comparisons and thus allow a better overview, especially when we compare five difference architectures (Table 1) and three hyper parameter sets (Table 2). This just made writing the analysis and discussions of the results easier. That is why we reported them first, followed by the qualitative results.
- The end points and limits of the proposed manuscript should be better assessed and defined.
We removed the misleading/wrong parts (localization is not future work) in the abstract and the conclusions as mentioned above in our response to your first comment. We hope that our changes clarified this confusion and thereby give a better definition of the end point and the limit of our manuscript.
Reviewer 2 Report
REVIEW of the manuscript titled:
sensors-1785797
Keypoint detection for injury identification during turkey husbandry using neural networks
Lameness is one of the most important causes of poor welfare in poultry, especially in turkey. The authors present a study that continues previous research, adding to it.
The title is concise, descriptive, and suggestive of the research conducted.
The abstract is a condensed version of the article and highlights the results obtained compared to the previous study.
The Introduction presents general information on the subject, and conclusions of the previous studies in correlation with the situation in the investigation area. The authors exploit a gap in previous research – unlike previous studies, this study focuses on a complete view of animals by identifying them with the help of top view cameras. The KPD model is combined with the segmentation model to locate the lesions.
Materials and method
Line 178-180: In order to better understand the way of working, the annotation of the key points and the delimitation by a rectangle of the turkeys, was it carried out during the growing period or at the end of the study?
Result and discussion
The authors use illustrations to clarify ideas and support conclusions.
Line 291-292: ‘’This could be increased even more with W48’’. I ask the authors to specify what they mean by this phrase?
Line 301-302: I ask the authors to justify the choice of the batch size of 64.
Line 318: Why was a different number of birds chosen for the quality assessment?
How do the authors think they can handle the probability of aggressive pecking in another area – for example at the cloaca, the lateral side of the wings, etc. How can it be counted?
The conclusions answer the problem posed in the Introduction. It also suggests future areas for research.
Although the study does not solve a major problem encountered in poultry farms, it makes an important contribution to monitoring injuries and sustaining animal welfare.
Author Response
Dear Reviewer 2.
Thank you for your constructive comments and your helpful input.
In the following you will find a list of your comments (in italics) and our reply (changes in the manuscript are marked in red).
- Line 178-180: In order to better understand the way of working, the annotation of the key points and the delimitation by a rectangle of the turkeys, was it carried out during the growing period or at the end of the study?
The annotations were conducted mainly on images showing turkeys in the mid and end of their fattening period. Basically there were also images of chicks and as well of injured ones (e.g. in the compartment for sick animals). In this study on keypoint detection, however, images of older birds were primarily used, as these images had the advantage that more animals could be seen in it, the birds were larger and therefore setting keypoints was easier (and hopefully more accurate). Furthermore, the occurrence of injuries also increased over the entire fattening period.
- Line 291-292: ‘’This could be increased even more with W48’’. I ask the authors to specify what they mean by this phrase?
Thank you for this hint. To clarify this, we rewrote the sentence. It now reads (L295-296):
This could be increased even more with the 48 layers of the HRNet-W48 network architecture.
- Line 301-302: I ask the authors to justify the choice of the batch size of 64.
A higher batch size allows the network to learn from more images at once and hence makes training more stable. However, a higher batch size also requires more GPU memory, 64 was the limit for our hardware. We added a short note to this effect. It reads (L304-306):
Our evaluations showed that the best model performance of HRNet-W48 was observed with a learning rate of 5e-4, a batch size of 64 (limited by available GPU memory), and 100 epochs of training (Table 2).
- Line 318: Why was a different number of birds chosen for the quality assessment?
We do not understand your question, as Line 318 did not present the number of birds. Could you please clarify your question?!
- How do the authors think they can handle the probability of aggressive pecking in another area – for example at the cloaca, the lateral side of the wings, etc. How can it be counted?
Thanks for that hint. Indeed, this is a limitation of our top-down view camera-based approach and we have added it to the discussion. It reads (L372ff):
One limitation of our top-down view camera-based approach is that certain affected regions, such as the cloaca or the lateral side of the wings, are hard to capture. Besides that, the annotation of keypoints on the all-white body of the turkeys was al-ready difficult, and thus we could not guarantee that, for instance, the ‘center of the body’ keypoint always had the same position.
Round 2
Reviewer 1 Report
The manuscript has been properly modified and assessed. It has been improved, no other changes seem necessary at this point.